# The Role of T Cells in Systemic Sclerosis: An Update

**Lazaros I. Sakkas \* and Dimitrios P. Bogdanos** 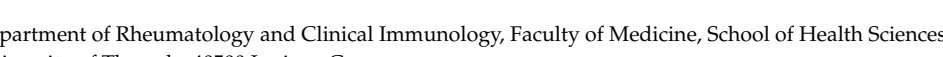

Department of Rheumatology and Clinical Immunology, Faculty of Medicine, School of Health Sciences, University of Thessaly, 40500 Larissa, Greece
\* Correspondence: lsakkas@med.uth.gr; Tel.: +30-241-3502813

**Abstract:** Systemic sclerosis (SSc) is a chronic disease characterized by microvasculopathy, autoantibodies (autoAbs), and fibrosis. The pathogenesis of the disease is incompletely understood. Microvasculopathy and autoAbs appear very early in the disease process. AutoAbs, such as those directed against DNA topoisomerase I (Topo I), are disease specific and associated with disease manifestations, and indicate activation of the adaptive immune system. B cells are involved in fibrosis in SSc. T cells are also involved in disease pathogenesis. T cells show signs of antigen-induced activation; T cells of TH2 type are increased and produce profibrotic cytokines interleukin (IL)-4, IL-13, and IL-31; CD4+ cytotoxic T lymphocytes are increased in skin lesions, and cause fibrosis and endothelial cell apoptosis; circulating T follicular helper (TFH) cells are increased in SSc produce IL-21 and promote plasmablast antibody production. On the other hand, regulatory T cells are impaired in SSc. These findings provide strong circumstantial evidence for T cell implication in SSc pathogenesis and encourage new T cell-directed therapeutic strategies for the disease.

**Keywords:** systemic sclerosis; T cells; T lymphocytes; Tregs; TH2 cells; treatment

## 1. Introduction

Systemic sclerosis (SSc) is a complex disease characterized by microvasculopathy with exaggerated response to stimuli and fibrointimal proliferation that leads to tissue ischemia, extensive fibrosis of skin and internal organs, and autoantibodies (autoAbs). The disease is divided according to the extent of clinical skin involvement into diffuse cutaneous SSc (dcSSc) and limited cutaneous SSc (lcSSc) and can be very difficult to manage leading to morbidity and increased mortality [1].

The pathogenesis of SSc is complex and incompletely understood. There is activation of adaptive and innate immune system and intermingled inflammatory, fibrotic, and vascular processes. Myofibroblasts, which lay down collagen and other extracellular matrix components, can be activated by cells of the immune system and derive from various cell types via trans-differentiation [2–4]. AutoAbs, including anti-Topoisomerase I abs (ATA, anti-Scl70), anticentromere abs, and anti-RNA polymerase III (ARP) antibodies, are detected very early, predict subsequent development of SSc, and indicate activation of the adaptive immune system [5]. Many autoAbs exert direct pro-fibrotic effects, and others also activate endothelial cells [6]. Furthermore, B cells with B cell antigen receptor (BCR) recognizing DNA-topoisomerase I with high affinity are increased in SSc, produce interleukin (IL)-6, and promote fibrosis in mice [7].

T cell involvement with production of profibrotic cytokines IL-4 and IL-13 is detected early before fibrosis in SSc [8]. Microvasculopathy also occurs very early in the disease process as endothelial cell apoptosis was detected in the early inflammatory stage of skin biopsies from SSc patients and even before any other histochemical dermal change in the University of California at Davis 200/206 chicken model of SSc [9].

Innate immune system signature, with toll-like receptors (TLRs) and type I interferons, is overexpressed in SSc [10,11]. In humans there are 10 different TLRs located in plasma membrane and endosomes, and they are expressed on immune and non-immune cells,

including fibroblasts, endothelial cells, and platelets. They recognize pathogen- and also danger-associated molecular patterns generated by tissue damage, and participate in SSc pathogenesis promoting fibrosis [10,12]. Type I interferons, IFNα and IFNβ, produced by innate immune cells, mainly plasmacytoid dendritic cells (pDCs) after TLR ligation, induce fibroblast collagen production [13]. pDCs were increased in skin and lung tissues in human SSc [14], whereas pDCs from healthy donors produce IFNα after stimulation with SSc sera containing anti-DNA-topoisomerase I autoAbs in a DNA/RNA manner [15]. pDCs also produce CXC ligand 4 (CXCL4, also known as platelet factor 4), a profibrotic and anti-angiogenic chemokine, and the most abundant chemokine in peripheral blood and skin in SSc patients, correlated with skin fibrosis, lung fibrosis, and pulmonary arterial hypertension (PAH), and predicted progression of SSc [16,17]

CXCL4 is an autoantigen for B cells in SSc [18] and forms complexes with self-DNA, which induce pDC IFNα production via endosomal TLR9 activation [19] and differentiation of B cells into antibody-secreting plasma cells [18]. Other cells, including macrophages and profibrotic growth factors, such as TGFb, also participate in the disease pathogenesis. Platelets also participate in SSc pathogenesis with the production of profibrotic, inflammatory, and vasoconstrictive mediators [20].

In this review, we examine the role of T cells in SSc, particularly recent research developments, and discuss how these findings relate to therapies. We searched PubMed using the terms systemic sclerosis, T cells, and T lymphocytes.

## 2. Antigenic Activation of T Cells in Systemic Sclerosis

The early presence of IgG autoAbs in SSc along with the presence of T cell infiltrates in skin before fibrosis and the production of T cell cytokines IL-4 and Il-13 indicate T cell activation very early in the disease process [8]. Next-generation sequencing of peripheral blood T cells in SSc revealed a profile associated with chronic antigenic stimulation [21,22]. Also, transcriptome analysis of peripheral blood T cell subsets in SSc revealed activated CD4+ and mucosal-associated invariant T(MAIT) cells (innate-like cells located at mucosal sites) but also increased expression of inhibitory molecule PD-1 which indicated functionally adapted T cells in response to chronic stimulation [22]. Increased frequency of T cells co-expressing inhibitory PD-1 and T immunoreceptor with immunoglobulin and ITIM domain (TIGIT) was also detected, suggesting an increased level of chronic exposure to antigenic stimulation and exhaustion [23].

Dendritic cells are the most potent antigen-presenting cells and are likely to function as such in SSc; pDCs' increase in SSc may also function as antigen-presenting cells [24]. In bleomycin-induced scleroderma mice, pDCs were increased in the affected organs, whereas pDC depletion reduced skin and lung fibrosis, as well as T cell and B cell infiltrates, and genes related to immune cell activation, chemotaxis, and fibrosis [14]. Macrophages and B cells have the machinery to process and present antigens to T cells and can function as antigen-presenting cells in SSc [25].

**T cell antigen receptor**. The conventional T cell antigen receptor (TCR), expressed on the vast majority of peripheral blood T cells, is composed of an α and a β chain. Sequence analysis of the β-chain of TCR of T cells from SSc skin lesions revealed oligoclonal T cell expansion and persistence of particular clones over 12 months [26]. Oligoclonal expansion of T cells in skin lesions from early SSc patients was also found by TCR complementarity-determining region 3 (CDR3) length analysis [27]. Another study reported oligoclonal expansion of peripheral blood CD4+ and CD8+ T cells from SSc that persisted over four years [28]. Also, CD4+CTLs (CD4+CD319+ T cells) strikingly increased in peripheral blood of SSc patients and very active in producing cytokines, were oligoclonal [29]. Maehara et al. [21] found a marked clonal expansion of CD4CTLs (CD28$^{low}$CD57$^{high}$CD4+) in peripheral blood from early dcSSc patients was associated with skin fibrosis. Furthermore, CD4+CTLs were cytotoxic to endothelial cells in early dcSSc skin lesions [21]. All these findings provide strong circumstantial evidence for an antigen-driven proliferation of T cells that contributes to disease fibrosis and vasculopathy.

The inciting antigen driving T cell proliferation is not known. The finding of the same T cells clones in skin lesions from early SSc patients and in co-cultures of SSc fibroblasts with autologous peripheral blood mononuclear cells (PBMCs) suggests that fibroblasts may provide autoantigens for T cell activation [27]. Endothelial cells, sharing B cell autoantigens with fibroblasts, may also provide autoantigens for T cells [30]. The detection of endothelial cell damage before fibrosis in skin biopsies supports this concept. Male offspring T cell clones generated from female SSc patients reacted with maternal HLA antigens and produced IL-4, raising the possibility for the contribution of GVHD mechanisms to the development of SSc [31]. Inorganic substances, such as hypochlorous acid (HOCL), can elicit an adaptive immune response, likely by modifying self-antigens. HOCL-induced mouse model of SSc induced by subcutaneous injection of HOCL, exhibit skin fibrosis with increased skin infiltration of CD4+ CD8+ T cells, macrophages, and B cells [32]. Interestingly, HOCL injections at first cause skin inflammation with highly proliferative T cells and myofibroblasts and later less inflammation and heavy collagen deposition, mirroring changes in human SSc [33].

## 3. T Cell Subsets and Function in Systemic Sclerosis

It has been found that skin changes with T cell infiltrates and endothelial cell apoptosis are very early events in the SSc process. T cells and macrophages were detected in the skin of SSc patients before histological evidence of fibrosis [34].

T cells are the predominant cell type in the inflammatory infiltrates of skin lesions in SSc [21,35] reaching 72 cells (mean number/7.36 mm$^2$) compared to 26 macrophages and to 5 B cells [35]. Next-generation RNA sequencing in skin biopsies from early dcSSc patients (disease duration 1.3 years) showed adaptive immune cell signatures to be associated with shorter disease duration [11]. T cells infiltrating skin lesions in SSc express the early activation antigen CD69 [36], a C-type lectin, and a marker of tissue-resident memory T cells which play a vital role in immune response and surveillance [37,38].

There is a complex interaction of T cells with B cells, macrophages, dendritic cells, fibroblasts, and endothelial cells which result in apoptosis or activation of endothelial cells and fibroblasts, and activation of B cells and macrophages with the production of autoAbs, profibrotic, inflammatory, and vasoconstrictive mediators, such as IL-4, IL-13, TGFβ, IL-6, CC chemokine ligand 2 (CCL2, also known as monocyte chemoattractant protein 1), endothelin, platelet-derived growth factor (PDGF) leading to fibrosis and microvasculopathy [8]. Animal models of SSc also support the role of T cells in the pathogenesis of fibrosis [39].

**TH2 cells**. T cells in peripheral blood in SSc are predominantly TH2 cells producing profibrotic cytokines IL-4 and IL-13 and TH2 cells are also detected in affected tissues [8,21,40,41]. In untreated early dcSSc skin among CD4+ T cells TH2 cells are detected along with TH17 cells, T follicular helper (TFH) cells, and regulatory T cells (Tregs) [21]. CD4+CCR7- memory T cells produced IL-13, IL-4, and TNFα, particularly in dcSSc [42]. Furthermore, CD8+ T cells overproducing IL-13 [41], and CD4+CD8+ double positive T cells overproducing IL-4, were detected in SSc skin lesions [43,44].

Apart from direct effect on fibroblasts, IL-4 stimulates macrophages to profibrotic alternatively activated phenotype (M2), which is mediated by IL-4 receptor (IL-4R), since blockade of IL-4Rα decreased alternatively activated (M2) macrophages and attenuated profibrotic changes in mice [45]. IL-4 signaling also stimulated proliferation of fibro/adipogenic progenitors in vitro [46]. IL-4 induces the development of granulocyte macrophage-colony stimulating factor (GM-CSF) producing CD30+B cells which are increased in SSc and promote differentiation of monocytes to profibrotic M2 macrophages [47]. Of note, GM-CSF induced trans-differentiation of monocytes to myofibroblasts [48]. A new TH2 cytokine, IL-31 [49], is detected at high levels in plasma, fibrotic skin and lung tissues [50], and skin fibroblasts from SSc patients [51]. IL-31 is a profibrotic cytokine as it induced genes related to cell proliferation and growth and suppressed genes related to angiogenesis in skin fibroblasts, and induced skin and lung fibrosis in mice [50]. Apart from direct effect on

fibroblasts, IL-31 promoted TH2 polarization in SSc, as it increased collagen production and expression of TGFβ and IL-6, IL-33, and CCL2 in SSc fibroblasts [51].

IL-4 derived from innate immune cells and the type of antigen can drive TH2 polarization in SSc [40]. Dendritic cells play a major role in T cell responses as they present antigens to T cells. Plasmacytoid dendritic cells (pDCs) appear to promote TH2 immune response and fibrosis. pDCs correlated with CD4+ T cells, IL-4-producing T cells, and IL-3 levels in the bronchoalveolar lavage (BAL) from SSc patients [14], whereas IL-3-stimulated pDCs induced TH2 differentiation [52]. CXCL4, a chemokine secreted by pDCs, induced endothelial cell activation and promoted TH2 cytokines in vitro [16].

IL-33, a member of IL-1 superfamily, widely expressed and released during cell damage [53], induces TH2 differentiation and is elevated in peripheral blood [54–56], affected skin and internal organs in SSc patients [57]. IL-33 enhanced IL-4-induced IL-31 production [49]. In mice, IL-33 induced IL-13-dependent cutaneous fibrosis [58,59]. IL-33 promoted extracellular matrix deposition and M2 macrophage polarization in diabetic mice [60].

**Cytotoxic T lymphocytes**. A recent study found that in the skin of untreated early dcSSc patients, the major component of inflammatory infiltrates was CD4+ cytotoxic (CD57$^{high}$CD4+) cells (CD4CTLs) and CD8+ T cells [21]. These CD4+CTLs apparently cause apoptosis of endothelial cells in SSc, as many apoptotic endothelial cells with granzyme B visible in their cytosol and in close proximity to CD4+CTLs expressed HLA-class II molecules in early dcSSc skin lesions [21]. Circulating CD28$^{low}$CD57$^{high}$CD4+ CTLs from early dcSSc patients were effector cells expressing genes related to metabolic activity, fibrosis, and cytotoxicity (granzyme B and perforin) and were associated with the number of myofibroblasts in paired skin samples [21]. These findings suggest that CD4+CTLs and CD8+ T cells are not only profibrotic but also contribute to microvasculopathy. Furthermore, the cytotoxic killing mediated by granzyme B and perforin can generate autoantigens that initiate and/or perpetuate immune response, as self-protein fragments generated by granzyme B were targets of autoAbs in SSc [61].

**Circulating T follicular helper cells**. T follicular helper (TFH) cells provide critical help for B cell differentiation and antibody production in germinal centers of secondary lymphoid organs [62,63]. They express inducible T cell costimulator (ICOS), programmed cell death protein 1 (PD)1, transcription factor BCL6, and produce IL-21 [62]. Circulating TFH cells (CD4+CXCR5+PD1+), increased in SSc and particularly in dcSSc, were activated and exhibited increased capacity to stimulate plasmablast secretion of IgG and IgM, partly mediated through IL-21R and JAK1/2 [64]. SSc patients with high TFH cells and plasmablasts more frequently progressed to late nailfold capillaroscopy pattern which correlated with internal organ involvement [65]. Another study found that a subtype of TFH cells (CXCR3+CCR6-) was increased and correlated with ACA, IL-21, and IL-6 serum levels in SSc [66]. ICOS+ TFH cells were also increased in the skin of SSc patients and in the skin of chronic graft-versus-host disease (cGVHD) mice and correlated with skin fibrosis whereas IL-21 neutralization decreased germinal center formation in cGVHD mice [67]. CXCL13+ TFH cells were adjacent to B cells in SSc skin biopsies [68]. All these findings suggest that TFH cells promote B cell autoAb responses in SSc.

**TH17 cells**. The role of Th17 cells in the pathogenesis of human SSc is unclear. IL-17A showed pro-fibrotic effects in animal models of SSc [69]. TH17 cells and IL-17A promoted skin and lung fibrosis [70], whereas IL-23, a survival and proliferation factor for TH17 cells, induced skin fibrosis in bleomycin-induced SSc mice [71], and IL-21, predominantly produced by TH17 cells and natural killer (NK) cells, inhibited the production of the anti-fibrotic T cell IFNγ [72]. The effect of IL-17 on fibrosis is less clear in human SSc [69]. Serum IL-17B, IL-17E, and IL-17F were elevated in SSc [73,74] and serum IL-21 levels were also elevated, particularly in dcSSc [75,76]. A recent study found anti-fibrotic effects of IL-17A. IL-17A induced apoptosis of SSc skin fibroblasts, whereas an upregulation of IL-17A associated with downregulation of collagen genes was found in co-cultures of peripheral blood mononuclear cells (PBMCs) from patients with early dcSSc and autologous skin

fibroblasts [77]. CD146+TH17 cells were increased in SSc and inversely correlated with lung fibrosis [78]. CD146 (also known as MUC18 or MCAM), a cell adhesion molecule expressed on TH17 cells and endothelial cells, is also detected as soluble CD146 (sCD146). CD146 is involved in fibrosis apparently through the Wnt pathway, since CD146 deficiency increased the susceptibility of mice to bleomycin-induced fibrosis and upregulated the canonical Wnt pathway [79].

**Angiogenic T cells**. Circulating angiogenic T (cTang) cells, which are CD3+ T cells characterized by the expression of platelet endothelial cell adhesion molecule-1 (CD31) and the receptor for the chemokine stromal cell-derived factor-1 (CXCR4, CD184) and involved in capillary formation, were increased in SSc particularly in SSc-associated PAH [80].

**Tregs**. Tregs are functionally impaired in SSc as they fail to produce inhibitory cytokines TGFβ1 and IL-10 and to suppress effector T cells [81,82]. This functional impairment is related to soluble factors, as plasma from early dcSSc patients completely abrogated the suppressive capacity of Tregs from healthy individuals [83]. Tregs ameliorated bleomycin-induced lung fibrosis, and similar effect was achieved by adoptive transfer of Tregs [84]. However, Tregs in affected SSc skin can produce profibrotic IL-4 and IL-13 cytokines [85] and this is likely due to the action IL-33, since IL-33 induced differentiation of Tregs into TH2-like cells producing IL-13 but not IL-4 [85]. Tregs may have a major role in protecting females from pulmonary hypertension (PH) and this may be relevant to SSc as PH is a serious manifestation of SSc. In animal models of PH, females with Tregs deficiency developed more severe PH, whereas Tregs co-cultured with human cardiac microvascular endothelial cells directly upregulated estrogen receptors and increased supernatant levels of prostacyclin and IL-10 [86].

**T follicular regulatory cells**. T follicular regulatory (TFR) cells, a subset of Tregs located within B cell follicles, co-express Treg and TFH markers and restrict TFH function [87]. IL-21 inhibits TFR differentiation [88]. Circulating TFRs (cTFRs) also exhibit the suppressive function of TFRs although one study reported that CXCR5+CD25+CD127-cTFRs lacked B cell suppressive activity [89]. However, data on TFRs in SSc are lacking. These findings suggest that T cells are involved in all aspects of SSc pathogenesis, namely fibrosis, vasculopathy, and autoAb production, and are summarized in Figure 1.

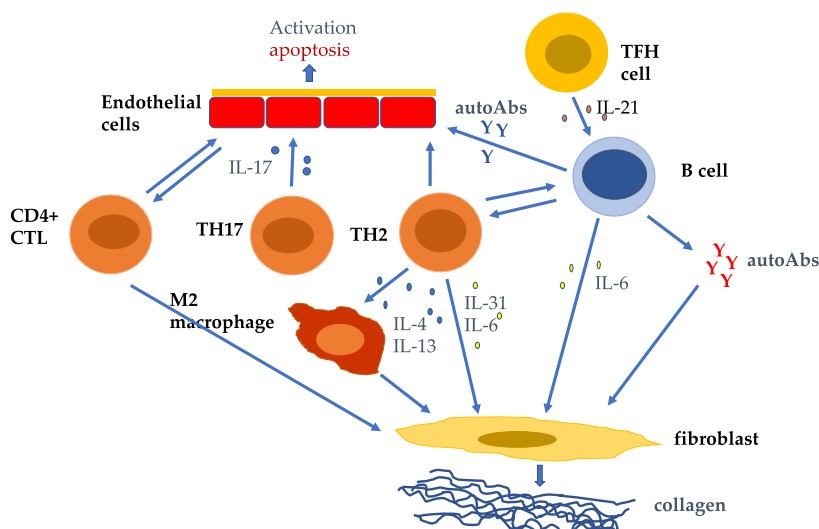

**Figure 1.** T cells are key players in the pathogenesis of systemic sclerosis (SSc) promoting autoAb production, fibrosis, and vasculopathy. TH2 cells and cytotoxic T lymphocytes (CTLs) predominate in SSc. TH2 cells producing IL-4, IL-13, IL-31, and IL-6 induce fibroblast collagen production. CD4+CTLs induce fibrosis and endothelial cell cellular cytotoxicity. TH17 cells induce endothelial cell adhesion molecules, chemokines, IL-1, and endothelial cell apoptosis [90]. TH2 and TFH cells help B cell production of autoAbs which activate fibroblast collagen synthesis and endothelial cell activation and apoptosis.

## 4. Therapeutic Implications

Immunosuppression is the standard of treatment of SSc today [91]. Steroids were positively associated with production of the antifibrotic IFNγ and negatively associated with production of profibrotic IL-4 and IL-13 by peripheral blood CD8+ T cells [92], but they are used with caution in SSc because high doses of steroids are associated with scleroderma renal crisis. Currently, autologous hematopoietic stem cell transplantation (AHSCT), the most effective treatment for severe SSc [93,94], is used to eliminate the autoimmune repertoire of SSc and replace it with a new immune system. Indeed, this concept is supported by the finding, at 1-year post AHSCT where the clinical improvement was related to renewal of the immune system with higher TCR diversity and newly generated Tregs and Bregs [95], as was also shown in multiple sclerosis [96]. Mycophenolate mofetil is an effective medication for skin tightness in SSc and is the standard-of-care for SSc-ILD stabilizing lung function [97]. Rituximab, a monoclonal antibody against CD20, expressed on B cells but also on a subset of T cells [98], improved skin and stabilized lung fibrosis in small observational studies in SSc [99], and the improvement of skin fibrosis was associated with decrease of CD4+ T cells in peripheral blood and skin [99,100].

Recent therapeutic efforts focused on particular immunopathogenic pathways. Abatacept, a cytotoxic T lymphocyte antigen 4(CTLA4) fusion protein, that binds CD80 and CD86 with high affinity and inhibits T cell activation and the antigen presentation to T cells by antigen-presenting cells [101], ameliorated established fibrosis in bleomycin-induced skin fibrosis in mice, and this was accompanied by a decrease in skin infiltrates of T cells, B cells, and macrophages [102]. Abatacept also ameliorated fibrosis in the cGVHD mouse model [102], and alleviated interstitial lung disease (ILD) and reversed PH in Fra-2 (Fos-related antigen-2) transgenic mice [103]. Of note, Fra-2 is part of the transcription factor activator protein(AP)-1, which is induced by stress signals [104]. Abatacept, in small observational studies in SSc [105,106] and in phase II RCT, showed some benefit although the decrease in modified Rodnan skin score (mRSS) did not reach statistical significance [107]. Importantly, Abatacept may have a beneficial effect on vasculopathy, since there was a decline of CD4+CTLs in SSc skin biopsies at 6-month post Abatacept [21]. Basiliximab, a chimeric moAb against CD25, the α chain of IL-2 receptor, showed efficacy in a case report and a small open-label study [108,109]. Neutralization of T cell profibrotic cytokines is an attractive strategy. Romilkimab, a bispecific monoclonal antibody against IL-4/IL-13, in a phase II study decreased Rodnan skin score in patients with early dcSSc [110].

Tocilizumab an anti-IL-6 receptor (IL-6R) moAb, which inhibits B cell differentiation, TH17 polarization, M2 macrophage polarization, and myofibroblast activation, reduced skin score and stabilized lung function in SSc patients [111,112].

JAK inhibitors may offer an attractive therapeutic choice. The JAK/STAT kinase pathways, present in T cells but also in fibroblasts and macrophages, are important downstream mediators of profibrotic responses, including IL-4, IL-13, and IL-6, and also type I interferons. In addition, JAK2 is a mediator of noncanonical TGFβ signaling in SSc fibroblasts [113,114]. A JAK1 inhibitor (itacitinib) and JAK2 inhibitor (TG101209), particularly the JAK2 inhibitor, decreased skin fibrosis and the JAK2 inhibitor decreased capillary loss in the bleomycin-induced skin fibrosis in mice, whereas a JAK3 inhibitor (JANEX1) did not have an effect on skin fibrosis [115]. Tofacitinib, a pan JAK inhibitor, decreased skin fibrosis and reduced skin IL-17 and TGFβ in bleomycin-treated mice [116]. Metformin, an antidiabetic agent which activates adenosine 5-monophosphate activated protein kinase (AMPK), inhibiting JAK/STAT pathway and mammalian target of rapamycin(mTOR), decreased skin fibrosis and reduced skin IL-17 and TGFβ in bleomycin-induced dermal fibrosis [116]. Ruxolitinib, a JAK1/2 inhibitor, suppressed the ability of SS cTFH cells to induce B cell proliferation and IgG production in vitro [64].

In a systematic review, Moriana et al. [117] summarized results from JAK inhibitors in SSc case series and case reports and demonstrated an efficacy in skin fibrosis. In a 52-week pilot study in human SSc, tofacitinib improved skin score more than methotrexate and led to healing of digital ulcers [118] whereas Baricitinib, an JAK1/2 inhibitor, in a small

number of SSc patients decreased skin score and led to complete healing of digital ulcers at week 24 [115]. Itacitinib, a JAK1 inhibitor (NCT04789850) and Belumosudil, an inhibitor of Rho-associated kinase2 (ROCK2) involved in STAT3/JAK pathway (NCT03919799 and NCT04680975), are currently studied in human SSc.

Imatinib, a tyrosine kinase inhibitor, decreased fibrosis in animal models of SSc [119]. In patients with SSc, imatinib reduced IL-4-producing T cells and pDCs in bronchoalveolar lavage [14] whereas a systematic review and meta-analysis showed some benefit on skin score but no clear efficacy on lung fibrosis [120].

Targeting TFH cells might be another therapeutic strategy for SSc [121]. Abatacept decreased TFH cells and TFH-dependent B cell hyperactivity in Sjogren's syndrome and rheumatoid arthritis [101]. Other TFH-directed agents may include anti-ICOS antibody which inhibited fibrosis and inflammation, and IL-21 neutralization which blocked disease progression and decreased TFH cell-induced plasma cell Ig production and germinal center formation in cGVHD mice [67]. In SLE patients IL-2 administration led to conversion of TFH cells to TFR cells [122] and this may be another strategy for SSc.

Re-enforcing Tregs could be another therapeutic strategy for SSc. Low-dose IL-2 restored the Tregs/TH17 effector cell balance [123] and decreased TFH cells associated with marked reduction in disease activity in SLE patients [124]. Various agents can induce Tregs. Retinoic acid, an active metabolite of vitamin A, induced CD4+CD25+FOXP3+ Tregs in SSc [125]. Dermal DCs producing aldehyde dehydrogenase, an enzyme that oxidizes retinaldehyde to retinoid acid, mediated induction of Tregs in bleomycin-induced skin fibrosis, whereas retinaldehyde dehydrogenase1 (RALDH1)+DCs was decreased in SSc skin lesions and inversely correlated with severity of skin fibrosis [126]. P-selectin glycoprotein ligand-1 (PSGL1) induced tolerogenic DCs with generation of Tregs [127], whereas deletion of PSGL1 promoted SSc changes. On the other hand, PSGL1−/− mice developed a SSc-like disease with dermal fibrosis, autoAb production, vascular damage [128], as well as PAH with reduction of Tregs and nitric oxide (NO) production [129]. Thalidomide, an agent contra-indicated in women of reproductive age, increased Tregs, reduced TH17 cells, and reduced skin and lung fibrosis in bleomycin-induced SSc mice [130].

Cenerimod, a selective inhibitor of sphingocin-1 phosphate 1 receptor, suppressed the infiltration of CD4+ T cells, CD8+ T cells, and macrophages, increased Tregs in the skin and spleen, and attenuated cutaneous and lung fibrosis in chronic GVHD mice, and attenuated bleomycin-induced cutaneous fibrosis [131].

An innovative therapeutic strategy could be the use of T cells with chimeric antigen receptor T cells (CAR-T cells) [132,133]. In CAR-T cells TCR is composed of the antigen-binding domain of heavy and light variable chains of a monoclonal antibody and, therefore, it recognizes its target in a non-HLA-restricted manner. CAR-T cells are successfully used in hematological malignancies. The use of CAR-T cells targeting pathogenic B cells, i.e., B cells recognizing DNA topoisomerase I with high affinity, targeting fibroblast activation protein, CAR natural killer cells that target PD-1 to eliminate TFH cells [134] or use of CAR-Tregs might be attractive strategies. Selected current and potential therapeutic agents are shown in Table 1.

**Table 1.** Selected therapeutic agents for systemic sclerosis.

| Agent | Action, Cells | Clinical Effects | Reference |
|---|---|---|---|
| ASCT | Renewal and reset of immune system. | Improves skin fibrosis and vasculopathy and stabilizes lung function. | [93–95] |
| Mycophenolate mofetil | Non-specific immunosuppressant of T cells and B cells. | Improves skin fibrosis and stabilizes lung function. | [91,97] |
| Cyclophosphamide | Non-specific immunosuppressant. | Stabilizes lung function. | [91] |

**Table 1.** *Cont.*

| Agent | Action, Cells | Clinical Effects | Reference |
|---|---|---|---|
| Rituximab | anti-CD20 moAb. Eliminates B cells, and a subset of T cells. | Improves skin fibrosis and stabilizes lung function. | [98–100] |
| Abatacept | CTLA4-Ig construct. Inhibits activated Tcells, and antigen-presenting cells. Decreases TFH cells and CD4+CTLs. | Improves skin score. | [21,105–107,135] |
| Basiliximab | Anti-CD25 moAb. Inhibits activated T cells. | Improves skin. | [108,109] |
| Romilkimab | Bispecific moAb against IL-4/IL-13 which are derived from TH2 cells, mast cells eosinophils, innate lymphoid cells. Inhibits M2 macrophages, and myofibroblasts. | Decreases skin score. | [110] |
| Anti-IL-21 moAb | Inhibits IL-21 which is produced by TH17 cells and TFH cells. Inhibits TH17 cells. Suppresses TFH cells. | Not studied in SSc. | [67] |
| Tocilizumab | moAb against IL-6R. Inhibits B cell differentiation, M2 macrophages, TH17 polarization, and myofibroblast activation. | Decreases skin score, stabilizes lung function. | [111,112] |
| Tofacitinib | Pan-JAK inhibitor. Inhibits activation of T cells, macrophages, and fibroblasts | Decreases skin score and promotes healing of DUs | [117,118] |
| Baricitinib | JAK1/2 inhibitor. Inhibits activation of T cells, macrophages, and fibroblasts. | Decreases skin score and promotes healing of DUs. | [115,117] |
| Ruxolitinib | JAK1/2 inhibitor. Inhibits activation of T cells, macrophages, and fibroblasts. Inhibits TFH cells. | Not studied in SSc. | [64] |
| Itacitinib | JAK2 inhibitor. Inhibits activation of T cells, macrophages, and fibroblasts. | Is being studied in SSc. | [115] |
| Low-dose IL-2 | Increases Tregs. Decreases TFH cells. | Not studied in SSc. | [123,124] |
| Thalidomide | Increases Tregs. Inhibits TH17 cells. | Not studied in SSc. | [130] |
| CAR-T cells | Against various selected cell targets. | Not studied in SSc. | [132–134] |

**Abbreviations:** ASCT: autologous stem cell transplantation; CAR-T cells: chimeric antigen receptor T cells; DUs: digital ulcers; JAK: Janus kinase; CTLA4: cytotoxic T lymphocyte associated protein 4; CD4+CTL: CD4+ cytotoxic T lymphocytes; TFH: T follicular helper cells.

## 5. Conclusions

T cells may be critical for SSc pathogenesis as they are involved in the development of fibrosis, autoantibodies, and vasculopathy, and recent evidence provides a rationale for innovative T cell-directed therapeutic strategies for this frequently devastating disease.

**Author Contributions:** L.I.S., conceptualization, L.I.S.; writing—original draft preparation, D.P.B.; writing—review and editing, L.I.S., artwork. All authors have read and agreed to the published version of the manuscript.

**Funding:** This research received no external funding.

**Institutional Review Board Statement:** Not applicable.

**Informed Consent Statement:** Not applicable.

**Data Availability Statement:** No new date generated from this study.

**Conflicts of Interest:** The authors declare no conflict of interest.

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
