# Peer review of "The Role of T Cells in Systemic Sclerosis: An Update"

_2673-5601, doi:10.3390/immuno2030034_

Round 1

Reviewer 1 Report

This review  article summarizes the role of T cells in Systemic Sclerosis (SSc) pathogenesis and highlights how recent evidence in this field may suggest the use of new therapeutic approaches in the treatment of this rare disease. 

The relevance of the topic is high. In fact, the SSc remains one of the most difficult rheumatological diseases to manage from a therapeutic point of view due to the absence of drugs that can modify its natural history. Therefore, evaluating new possible therapeutic targets is essential in this research field.

The review is clear  and It's based on appropriate references that cover the most important advances in T cells research field.

These are my comments:

1. Jak inhibitors have a very important role in blocking cellular activation mediated by cytokines. The authors had already mentioned in a previous paragraph the role of JAK1 / 2 in the activation of THF cells (row 115, reference 42).  Given the recent use of this category of drugs in SSc, why did the authors not mention them in paragraph 4 "Therapeutic implications"? 

2. The figure in the text is not numbered and the title is placed above without a caption that can summarize the role of T cells in SSc pathogenesis. Moreover it would be very useful, if possible, to indicate on which pathway the mentioned drugs works on.

3. Row 96 check bold characters (i of in)

4. Row 270 : please check typo (conseptualization)

Author Response

We thank you for your efforts and your valuable comments. We made recommended changes addressing Reviewers’ comments, which are underline in the text. We hope that you will find satisfactory these changes. 

1.We added the JAK inhibitors in the therapeutics section and the added Table

  1. We amended the Figure. We added cytokines but not medications since we added a Table with medications

3, 4. We corrected the typographical errors

Reviewer 2 Report

The work is essentially based on the involvement of the adaptive immune system involved in scleroderma (SSc) but in order for an adaptive response to be activated, first of all there is an abnormal innate immune response that is responsible for the initial events of the disease. The dysfunction and activation of plasmacytoid dendritic cells (pDc) with the involvement of TLR 7/9 and the Type I Interferon Signature are common aspects in the initial stages of scleroderma as well as the activation of myeloid dendritic cells (mDc) via TLR 7/8. Activation of TLRs on dendritic cells through the adjuvant action of CXCL4, able to associate with self DNA/RNA, released by apoptotic cells is one of the mechanisms that explains how self tolerance is bay-passed. CXCL4–DNA/RNA complexes induce IFN-α in pDCs and direct B-cell stimulation (by TLR expressed by B cells), including the secretion of anti-CXCL4 antibodies. Anti-CXCL4 antibodies may further increase pDC stimulation and IFN-α release in vivo, creating a vicious cycle which sustains the SSc IFN-I signature and general inflammation. The interaction of innate and adaptive immune cells appears crucial in SSc. There are some examples of typical SSc autoantibodies that activate innate immune cells, fuel inflammation and can contribute to the IFN-I signature and fibrosis. For instance, Kim’s group demonstrated that SSc sera containing autoantibodies that mark SSc, such as anti-centromere (ACA) and anti-topoisomerase (ATA) antibodies, induced high levels of IFN-α in healthy donor (HD) peripheral blood mononuclear cells (PBMCs) in a pDC- and RNA/DNA-dependent manner. A paper by Eloranta et al. also showed that sera from SSc patients, mixed with necrotic/apoptotic material, induced IFN-α production in pDCs, apparently activated by immunoglobulin (Ig) immune complexes (ICs) formed by SSc autoantibodies. Interestingly, SSc IgG immune complexes activated nucleic acid responsive TLRs (TLR7/8/9), suggesting that extracellular release of nucleic acids and endosomal TLRs are important in SSc pathogenesis. The ability of CXCL4 to bind cationic molecules including DNA, in the form of crystals, suggested that anti-CXCL4 antibodies could be easily generated in SSc. Indeed, formation of particulate structures, such as large molecular complexes and/or crystal structures containing self-molecules, confers greater antigenicity to otherwise less immunogenic proteins. The data reveals that anti-CXCL4 antibody autoreactivity is present in a consistent proportion of SSc patients and correlates with IFN-α, while T-cells also recognized CXCL4 as an autoantigen, suggesting T-cell help for autoantibody production. Culture of  B-cells initially CD19pos/CD27pos (namely memory B-cells) and CD38low, with CXCL4–DNA immune complexes, are able to differentiate into plasma cells in vitro. Secretion of total IgG tended to be significantly higher in the presence of CXCL4–DNA or –RNA  immune complexes . Thus, the effect was likely mediated via TLR9, and possibly TLR7 stimulation. CXCL4–RNA complexes were also able to induce these effects in vitro. However, it is also known that the production of high affinity antibodies is ensured by helper  CD4 T-cells. , which indicates that CXCL4 is acting as a novel autoantigen in SSc and reinforce the assumption that CXCL4 can bind polyanionic compounds to become immunogenic. So new therapeutic approch are  considering also this aspect of SSc and CXCL4 involvement. Finally, NETosis is also involved in stimulation of innate immune system. The authors don't consider  pro-fibrotic factors such as endothelin and CTCF, as such as the production of Ros species that can account respectively on one hand of  fibrosis and  on the other side of generation of neo antigens from self molecules.  All these pro-fibrotic factors are generated by immune cells located at the site of cell damage and inflammation. Other therapy also consider these factors (for example iloprost). Finally transition of endothelial cells to mesenchymal ones is another important event at beginning of endothelial cells damage and dermal  fibrosis. I thinks that all these aspects had to be exposes also briefly in this work to give a more complete knockout. A searcher by PubMed using the terms : systemic sclerosis , T cell and T lymphocytes, surely it will have also highlighted these aspects, important  for new therapies

Author Response

We thank you for your efforts and your valuable comments. We made recommended changes addressing Reviewers’ comments, which are underline in the text. We hope that you will find satisfactory these changes

We added a paragraph on pDCs type I interferons CXCL4 and TLRs, as suggested, in the introduction section of the manuscript. We feel that a long overview of the pathogenesis of SSc will dilute our focus on T cells

Reviewer 3 Report

The review paper "the role of T cells in systemic sclerosis: An update" brought together our current unedrstanding of the role of t cells in Systemic sclerosis (SSc).

The paper covers some essential findings. However, the structure needs to be re-shaped, and more information is required to fill the gaps in the text.   

1- Page 2, LINE 48: there are more important findings regarding the roles of T cells. The ref 19 can come later when you are talking about infiltration. To start the T cell section.
Consider changing the OPENING sentence. T cell activation is the first step in the T cell story. Also, because DC cells (or perhaps B cells) come first in the antigen presentation, you may first (briefly) discuss their role in antigen presentation to T cells and then shape up the story by T cells.

 2- It's recommended to start with differentiation, followed by activation and function, cross-talk with other cell types, and at the end, look for in vivo data (animal study). This is up to you, but it is always an excellent approach to be consistent when discussing some relative subsets.

3- You must also look at the role(s) of T follicular regulatory (Tfr) cells. There is some evidence that they work closely with Tfh cells and control the antibody response, which may lead to inflammation.

4- "Antigenic activation of T cells in systemic sclerosis" (LINE 160) must be discussed before talking about T cell subsets. As mentioned in comment 1 (above), DCs comes first in the antigen presentation and then we can continue the story with TCR activation.

5- at least one more figure is required to show the story of Tregs, Tfh/Tfr and Th17 cells.

6- For the therapeutic section (LINE 202), it's recommended to use a table outlining the cell type, name and effect of the medicine. It's essentially important for clinicians to see how the basic research helps the clinic and vice versa.

Good luck

Author Response

We thank you for your efforts and your valuable comments. We made recommended changes addressing Reviewers’ comments, which are underline in the text. We hope that you will find satisfactory these changes

1.We modified the beginning of T cell section.

2.We added a paragraph on Tcell cross-talk with other cells and animal studies

3.We added a small section on TFRs

4.We moved the antigenic activation of T cells before the Tcell subsets and function section, as suggested and discussed cross-talk with other cells within each Tcell subset section

  1. Since the role of TH17 cells in human SSc may be bidirectional we felt that we shouldn’t put too much emphasis on TH17 cells. Besides, there no data on TFRs in SSc. For these reasons we did not include a Fig on TH17 Tregs TFRs
  2. We added a Table in the therapeutic section, as suggested.

Round 2

Reviewer 3 Report

The paper is now aceptable.